# The Unique Lipidomic Signatures of *Saccharina latissima* Can Be Used to Pinpoint Their Geographic Origin

**DOI:** 10.3390/biom10010107

**Published:** 2020-01-08

**Authors:** João P. Monteiro, Felisa Rey, Tânia Melo, Ana S. P. Moreira, Jean-François Arbona, Jorunn Skjermo, Silje Forbord, Jon Funderud, Diogo Raposo, Philip D. Kerrison, Marie-Mathilde Perrineau, Claire Gachon, Pedro Domingues, Ricardo Calado, M. Rosário Domingues

**Affiliations:** 1Centro de Espetrometria de Massa, Departamento de Química & QOPNA, Universidade de Aveiro, Campus Universitário de Santiago, 3810-193 Aveiro, Portugal; felisa.rey@gmail.com (F.R.); taniamelo@ua.pt (T.M.); ana.moreira@ua.pt (A.S.P.M.); p.domingues@ua.pt (P.D.); mrd@ua.pt (M.R.D.); 2Departamento de Química & CESAM & ECOMARE, Universidade de Aveiro, Campus Universitário de Santiago, 3810-193 Aveiro, Portugal; 3Departamento de Biologia & CESAM & ECOMARE, Universidade de Aveiro, Campus Universitário de Santiago, 3810-193 Aveiro, Portugal; rjcalado@ua.pt; 4C-Weed Aquaculture, 35400 St. Malo, France; info@c-weed-aquaculture.com; 5Department of Environment and New Resources, SINTEF Ocean, 7465 Trondheim, Norway; jorunn.skjermo@sintef.no (J.S.); silje.forbord@sintef.no (S.F.); 6Seaweed Energy Solution AS, Bynesveien 48, 7018 Trondheim, Norway; funderud@seaweedenergysolutions.com (J.F.); raposo@seaweedenergysolutions.com (D.R.); 7Scottish Association for Marine Science, Oban PA37 1QA, UK; philip.kerrison@sams.ac.uk (P.D.K.); marie-mathilde.perrineau@sams.ac.uk (M.-M.P.); claire.gachon@sams.ac.uk (C.G.)

**Keywords:** elemental composition, glycolipids, lipidomics, mass spectrometry, phospholipids, polyunsaturated fatty acids, seaweeds, traceability

## Abstract

The aquaculture of macroalgae for human consumption and other high-end applications is experiencing unprecedented development in European countries, with the brown algae *Saccharina latissima* being the flag species. However, environmental conditions in open sea culture sites are often unique, which may impact the biochemical composition of cultured macroalgae. The present study compared the elemental compositions (CHNS), fatty acid profiles, and lipidomes of *S. latissima* originating from three distinct locations (France, Norway, and the United Kingdom). Significant differences were found in the elemental composition, with Norwegian samples displaying twice the lipid content of the others, and significantly less protein (2.6%, while French and UK samples contained 6.3% and 9.1%, respectively). The fatty acid profiles also differed considerably, with UK samples displaying a lower content of *n*-3 fatty acids (21.6%), resulting in a higher *n*-6/*n*-3 ratio. Regarding the lipidomic profile, samples from France were enriched in lyso lipids, while those from Norway displayed a particular signature of phosphatidylglycerol, phosphatidylinositol, and phosphatidylcholine. Samples from the UK featured higher levels of phosphatidylethanolamine and, in general, a lower content of galactolipids. These differences highlight the influence of site-specific environmental conditions in the shaping of macroalgae biochemical phenotypes and nutritional value. It is also important to highlight that differences recorded in the lipidome of *S. latissima* make it possible to pinpoint specific lipid species that are likely to represent origin biomarkers. This finding is relevant for future applications in the field of geographic origin traceability and food control.

## 1. Introduction

Increased pressure raised by demographic demands and otherwise restricted resources has motivated the world to turn its eyes to the sea in the search for answers and sustainable solutions. In this context, macroalgae have emerged as potential resources to be explored not only because of their recognized nutritional value, but also because of some new high-end uses and applications that have emerged in recent decades. Seaweeds are valuable nutritional sources of proteins, carbohydrates, minerals, vitamins, and polyunsaturated fatty acids (PUFA) [1]. Macroalgae are also valuable sources of bioactive and industrially interesting compounds including pigments, lipids, fatty acids, polysaccharides, phenolics, lectins, alkaloids, terpenoids, and halogenated compounds of interest to the food, pharmaceutical, cosmetic, nutraceutical, and biomedicine industries [2,3]. Therefore, aquaculture production of macroalgae is justifiably prospering [4].

*Saccharina latissima* is an edible brown seaweed widely distributed in shallow-water habitats in the North Atlantic, North Pacific, and the Arctic [5]. Aquaculture of this species can give high biomass yields, e.g., 150–200 tons per hectare per year off the coast of Norway, and several commercial seaweed farms have been established mainly for its production [6]. *Saccharina latissima* has been shown to have a high content of essential minerals (such as magnesium, iodine, and calcium), sugars (for instance glucose and mannitol), proteins, and other valuable compounds such as phenolics (especially phlorotannins), fucoidan, fucoxanthin, and alginates [7,8,9,10,11].

Several works have already targeted the lipid content of *S. latissima* and characterized its fatty acid composition, revealing the presence of important lipid species, including a considerable percentage of PUFA and particularly of *n*-3 fatty acids [12,13,14,15]. The *Saccharina latissima* polar lipidome has already been described, revealing the existence of 197 different molecular species of polar lipids, including glycolipids, phospholipids, and betaine lipids, for some of which important bioactive properties have been proposed [13].

Abiotic and biotic factors have been proposed to effectively influence not only yield but also the biochemical composition of macroalgae [3,16]. Indeed, macroalgae have been shown to react to environmental changes by shifting their nutrient uptake efficiency, photosynthetic activity, and secondary metabolism [17,18], with obvious repercussions in compositional terms. The physicochemical environmental factors known to impact macroalgae development and chemical composition include light, temperature, salinity, CO_2_ concentration, nutrient availability, the presence of contaminants, and biotic interactions [16,19,20,21]. Most of these factors impacting composition are bound to vary geographically, creating dissimilar growth conditions. In the case of *S. latissima*, biomass yield, morphology, and biochemical composition have already been reported to be influenced by growth location [8,22,23]. Local idiosyncratic singularities in terms of compounds with particular nutritional or commercial value may represent new marketing and economic opportunities. In this sense, this study will allow the potential macroalgae to be maximized and valorization of specific production sites.

Therefore, in this work, we used state-of-the-art high-resolution LC-MS-based lipidomics approaches to analyze origin-driven idiosyncrasies at the molecular level. This approach was able to produce evidence of differentiation between geographical origins with remarkable resolution, which is interesting not only from nutritional and pharmacological standpoints but also may represent a useful tool with which to trace the geographic origins of macroalgae.

## 2. Materials and Methods 

### 2.1. Sampling

Samples of *S. latissima* from France were cultivated by C-Weed Aquaculture in the vicinity of St. Malo (N48.4351; W4.0163, St. Malo, France) and collected on the 29th of May 2017. Samples from Norway were cultivated at Seaweed Energy Solutions’ farm Taraskjæra (N63.4228, E8.5223, Frøya, Norway) and harvested on the 28th of April 2017. United Kingdom samples were cdultivated at the Scottish Association for Marine Science´s farm (“Port A’Bhuiltin farm” or “Lismore farm”, N56.4883, E5.47133, Oban, UK) and harvested on the 24–26th of May 2017. Upon harvesting seaweeds were hung for a few minutes to dry, removing as much as seawater as possible, and then stored at −20 °C Whole seaweed portions from each location were freeze-dried, grinded with a mortar and pestle and kept at −80 °C before analysis. For each localization, five different random portions of the biomass were used.

### 2.2. Biochemical and Elemental Compositional Analysis 

For moisture determination, algal biomass (250 mg) was weighed into ceramic crucibles and dried overnight in a drying oven (105 °C, for 16 h). Crucibles were then cooled to room temperature and weighed to determine water content. Afterwards, ash determination was performed on the same biomass samples. First, biomass was pre-incinerated in crucibles by maintaining the crucibles on a heated plate for 20 min. Crucibles were then transferred to a muffle furnace and kept at 575 °C for 6 h. After cooling to room temperature, the crucibles were weighed and the ash content determined.

The elemental composition of the biomass regarding carbon (C), hydrogen (H), nitrogen (N), and sulfur (S) was evaluated using a Leco Truspec-Micro CHNS 630-200-200 elemental analyser at a combustion furnace temperature of 1075 °C and afterburner temperature of 850 °C. Portions of about 2 mg of macroalgal biomass were burned in an oxygen/carrier gas mixture, strictly following all conditions guaranteeing full combustion and the conversion of a few byproducts to water vapor, carbon dioxide and nitrogen for gas analysis. Carbon, hydrogen and sulfur were detected using infrared absorption, and nitrogen using thermal conductivity. The elemental analysis allowed estimation of the protein content by the use of a nitrogen-to-protein conversion factor. Although a factor of 6.25 is normally used, here we used a reportedly more suitable factor of 5, which has been proposed as a universal nitrogen-to-protein conversion factor for seaweeds [24]. The carbohydrate (and other compounds) percentage was estimated by subtracting the percentage of ash, lipids, and proteins in samples from 100%.

### 2.3. Lipid Extraction

*Saccharina latissima* total lipid extracts were obtained using a modified version of the Bligh and Dyer method [13,25,26]. Briefly, a total biomass of 250 mg was mixed with 2.5 mL of methanol and 1.25 mL of chloroform in glass centrifuge tubes, homogenized, briefly sonicated, and incubated on ice for 2 h and 30 min on an orbital shaker. After centrifugation at 626× *g* for 10 minutes at room temperature, the organic phase was collected. Re-extraction steps were performed using the original biomass, and the resulting organic phases combined with the first. Water was added to the organic phase in order to resolve a two-phase system and, after another centrifugation step in the same conditions as before, the lipid-containing organic lower phase was collected. This final fraction was dried under a nitrogen stream, the weight of the extracts was determined by gravimetry, and the extracts were preserved at −20 °C. 

### 2.4. Fatty Acid Analysis Using Gas Chromatography–Mass Spectrometry (GC-MS)

Total fatty acids of lipid extracts were analyzed using GC-MS upon transmethylation [13,25] by derivatizing aliquots of 30 μg of each lipid extract dissolved in 1 mL of *n*-hexane containing a C19:0 internal standard (0.75 μg mL^−1^, CAS number 1731-94-8, Merck, Darmstadt, Germany). The resulting fatty acid methyl esters (FAMEs) were dissolved in 50 µL *n*-hexane, and 2 μL of this solution was injected into an Agilent Technologies 6890 N Network Chromatograph (Santa Clara, CA, USA) equipped with a DB-FFAP column with a length of 30 m, an internal diameter of 0.32 mm and a film thickness of 0.25 μm (J&W Scientific, Folsom, CA, USA) connected to an Agilent 5973 Network Mass Selective Detector. Operating settings used have been described previously [13]. Fatty acid identification was performed by comparing the retention times and mass spectra obtained to those of the commercial FAME standards in the Supelco 37 Component FAME Mix (ref. 47885-U, Sigma-Aldrich, Darmstadt, Germany). The average chain length (ACL), double bond index (DBI), peroxidizability index (PI), and content of monounsaturated fatty acids (MUFA), polyunsaturated fatty acids (PUFA), polyunsaturated fatty acids *n*-3 (PUFA *n*-3), and polyunsaturated fatty acids *n*-6 (PUFA *n*-6) were determined as previously described [27].

### 2.5. Hydrophilic Interaction Liquid Chromatography–Mass Spectrometry (HILIC-LC-MS)

Total lipid extracts were analyzed by hydrophilic interaction liquid chromatography using a high-performance liquid chromatography (HPLC) Ultimate 3000 Dionex (Thermo Fisher Scientific, Bremen, Germany) with an autosampler coupled online to a Q-Exactive hybrid quadrupole mass spectrometer (Thermo Fisher, Scientific, Bremen, Germany). The solvent system and instrumental settings were set as described previously [28]. To perform the HILIC-LC-MS analyses, 10 µg of total lipid extract, 2 µL of phospholipid standards mix (0.01 µg dimyristoylphosphatidylcholine (dMPC), 0.01 µg dimyristoylphosphatidylethanolamine (dMPE), lysophosphatidylcholine (LPC) 0.01 µg, 0.04 µg dipalmitoylphosphatidylinositol (dPPI), 0.006 µg dimyristoylphosphatidylglycerol (dMPG), 0.02 µg dimyristoylphosphatidylserine (dMPS), 0.04 µg tetramyristoylcardiolipin (tMCL), 0.01 µg sphingomyelin (SM(17:0/d18:1)), 0.04 µg dimyristoylphosphatidic acid (dMPA)) and 88 µL of eluent (40% of mobile phase A and 60% of mobile phase B) were mixed and injected into the Ascentis Si column HPLC Pore column (15 cm × 1 mm, 3 µm, Sigma-Aldrich, St Louis, USA), with a flow rate of 40 µL minutes^−1^ at 30 °C. Acquisition in the Orbitrap® mass spectrometer was performed in both positive (electrospray voltage 3.0 kV, Thermo Scientific, Waltham, USA) and negative (electrospray voltage −2.7 kV) modes. For lipidomic analysis, phospholipid peak integration and assignments were performed using MZmine version 2.32 (Boston, USA) [29]. For all assignments, ions within 5 ppm of the lipid exact mass were considered. Analysis of the MS/MS spectra acquired in the positive ion mode was performed to confirm the identity of the molecular species belonging to the MGMG, DGMG, MGDG, DGDG, DGTS, PC and LPC classes. The MS/MS spectra acquired in the negative ion mode were used to confirm the identity of SQDG, SQMG, LPE, PE, LPG, PG, LPI, PI, lysophosphatidic acids (LPA) and phosphatidic acids (PA). Negative ion mode MS/MS data were used to identify the fatty acid carboxylate anion fragments RCOO^−^, which allowed the assignment of the fatty acyl chains esterified to the PL precursor. All the ions detected and MS/MS fragmentation patterns characteristic of the lipid classes detected and analyzed in the present study, acquired both in positive and negative ion modes, are available online as Appendix A. Normalization of the data was performed by dividing the peak areas of the extracted ion chromatograms (XICs) of the polar lipid precursors of each class (listed in Appendix A) by the peak area of the internal standard selected for the class.

### 2.6. Statistical Analysis

Multivariate and univariate analyses were performed using R version 3.5.1 [30] in Rstudio version 1.1.4 [31]. The statistical significance of differences among samples from different origins (FR, NO and the UK) was assessed by Kruskal–Wallis test, followed by Dunn’s multiple comparison test with Benjamini and Hochberg FDR correction (*q* values). Differences with *q* value < 0.05 were considered statistically significant. All experimental data are the result of the analysis of five replicates per origin). GC data were glog transformed and HPLC/MS data were glog transformed and autoscaled using the R package Metaboanalyst [32]. Principal component analysis (PCA) was carried out with the R built-in function and ellipses were drawn using the R package ellipse [33], assuming a multivariate normal distribution and a level of 0.95. Hierarchical clustering heatmaps were created with the R package pheatmap [34] using "Euclidean" as clustering distance, and "ward.D" as the clustering method. Graphics and boxplots were created using the R package ggplot2 [35]. Other R packages used included plyr [36], dplyr [37], and tidyr [38].

## 3. Results

### 3.1. Biochemical and Elemental Compositional Analysis 

For the biochemical compositional we estimated the moisture, ash, lipid, protein, and carbohydrate contents. Principal component analysis (PCA) for the compositional parameters evaluated showed that replicates clustered clearly according to geographical origin (Figure 1A). PCA analysis described 97.6% of the total variance, and Principal Component 1 described most of this variance (94.9%). The parameters contributing most to the discrimination between production sites are ranked in Figure 1B. *Saccharina latissima* from the UK had the highest moisture percentage, while the Norwegian samples presented the lowest (*q* = 0.001, for NO vs. UK; Figure 1B). Nevertheless, since differences in moisture content between locations may have been a result of different conditions and preparation methods at each sampling location, the following calculations and results were made in relation to dry weight, so these differences were not transposed to the later findings. Regarding ash percentages, all samples presented significant inorganic content, with Norwegian biomass displaying significantly higher values than that from the UK (*q* = 0.001; Figure 1B). In our determinations, Norway’s *S. latissima* presented a considerably higher lipid content than France’s (*q* = 0.001), considering the estimation of the lipid percentage per dry biomass (Figure 1B). In terms of lipid percentages regarding total organic content (lipid content divided by total lipids plus proteins and carbohydrates), Norway samples displayed a higher content (4.45% ± 0.16), while French and UK samples had lower contents (1.90% ± 0.14 and 1.63% ± 0.17, respectively). The estimated carbohydrate content was higher in *S. latissima* from the UK and lower in that from Norway (*q* = 0.001, for NO vs. UK; Figure 1B). Regarding the elemental composition, differences were also found in carbon, hydrogen, sulfur, and nitrogen composition (Figure 1B). The most striking differences in the elemental composition concerned the high sulfur content and lower nitrogen percentage/protein levels in Norwegian seaweed compared to the samples from the UK (*q* = 0.002 and *q* = 0.001, respectively). 

### 3.2. Fatty Acid Composition

The fatty acid profile in the *S. latissima* samples from different geographical origins was determined by GC-MS. Palmitic acid (16:0) was the main fatty acid present in all three cultivation sites (Table 1). Oleic acid (18:1 *n*-9) was the second most prevalent fatty acid in Norwegian and UK samples, but not in French samples, in which eicosapentaenoic acid (20:5 *n*-3, EPA) was the second most abundant fatty acid. From this point, the fatty acid profiles varied greatly, exposing a high sensitivity of fatty acid content to the cultivation site. However, all samples of *S. latissima* showed an overall high percentage of PUFAs, and generally displayed significant content of essential fatty acids, namely 18:2 *n*-6, 18:3 *n*-3, 20:4 *n*-6, and 20:5 *n*-3, similarly to what was previously reported in the literature [12,13,14,15]. PCA for fatty acid composition showed that replicates were clearly clustered according to geographical origin in a two-dimensional score plot representing the analyses (Figure 2A) and describing 76.7% of the total variance, including Principal Component 1 (60.6%) and Principal Component 2 (16.1%). PUFAs were the most relevant fatty acids contributing to the discrimination between geographical growth locations. The fatty acids 18:4 *n*-3, 16:0, 22:0, and 18:3 *n*-3 were the major contributors to the variability observed along the first dimension of the PCA (see Figure 2B). The fatty acid profiles were further analyzed and compared by calculating parameters of nutritional and functional interest (Table 1). *Saccharina latissima* from Norway displayed a higher content of PUFAs along with an increased double bond index (DBI) compared to UK biomass (*q* = 0.007, and *q* = 0.018, respectively). Another parameter providing insight about membrane biophysical properties, the average chain length (ACL), was not changed by geographical culture location. 

### 3.3. Polar Lipidome Analysis by HILIC-LC-MS

Polar lipids from samples of *S. latissima* from the three growth locations were analyzed in total lipid extracts using modern lipidomic approaches based on high-resolution LC-MS. A total of 217 molecular lipid species were detected, belonging to three principal classes of polar lipids, namely glycolipids (41 galactolipid and 36 sulfolipid molecular species), phospholipids (127 molecular species), and betaine lipids (13 molecular species), similar to the results of a previous work characterizing the polar lipidome of *S. latissima* from Norway [13]. A complete list of the lipid species detected is provided in the Appendix A. With regard to the previous study on the polar lipidome of *S. latissima* from Norway [13], we found a similar profile in our samples from the same origin.

The samples from different origins were found to differ in some of the lipid molecular species present, but also in their amounts. In *S. latissima* from France, 216 molecular species were identified, while 193 were identified in samples from Norway and 208 in samples from the UK (Appendix A). In the French biomass, 41 different molecular species of galactolipids were found. Norwegian and UK samples presented a lower number of different galactolipid species (36 and 39, respectively). *S. latissima* produced in Norway displayed a minimal content of lysophosphatidylethanolamines (LPE), while French samples contained one fewer phosphatidylethanolamine (PE) species (the only species found in the others that was not present in samples from France; Appendix A; Figure 3B). Five lysophosphatidylglycerols (LPG) and 16 phosphatidylglycerol (PG) species (one less in the UK; Appendix A) were detected. Regarding lipids belonging to the phosphatidylinositol (PI) class, 2 lysophosphatidylinositols (LPI) and 16 PI molecular species were present (3 fewer in Norway; Appendix A). Norwegian seaweed displayed less variability in betaine lipid (DGTS) species (Appendix A). 

Statistical analysis was performed taking into account the total lipid per class, calculated as the sum of the normalized (to internal standard) peak areas of each molecular species belonging to that class (Figure 3), as well as considering all the individual lipid species identified and quantified in the total lipidome (Figure 4).

PCA analysis for lipid classes (sum of normalized peaks for each lipid class) demonstrated evident dissimilarities between different growth locations (Figure 3A), forming a two-dimensional score plot representing 90.3% of the total variance, including Principal Component 1 (70.3%) and Principal Component 2 (20.0%) (Figure 3A).

PCA loading values showed that the more discriminating classes were monogalactosyldiacylglycerols (MGDG), lysophosphatidylcholines (LPC), sulfoquinovosyl monoacylglycerols (SQMG), and digalactosylmonoacylglycerols (DGMG) (Figure 3B), which essentially set France apart from the other production locations. Norwegian and UK samples were mainly discriminated by their total contents in PI, phosphatidylcholines (PC), LPE, and digalactosyldiacylglycerols (DGDG), as can be seen in the box plots presented in Figure 3B.

French samples were more abundant in DGMG and monogalactosylmonoacylglycerol (MGMG) species than those from the UK (*q* = 0.001 for both comparisons), and less abundant in MGDG than samples from Norway (*q* = 0.001; Appendix A; Figure 3B). Norwegian seaweed had a higher DGDG content than that from the UK (*q* = 0.002; Figure 3B). *Saccharina latissima* from France presented a remarkably higher content of SQMG (*q* = 0.011 for FR vs. NO, *q* = 0.024 for FR vs. UK; Appendix A; Figure 3B). French biomass also had a remarkably higher content of LPCs than Norway’s (*q* = 0.001; Figure 3B).

Statistical analysis was also performed considering the semi-quantification of the 217 lipid species identified. Multivariate analysis by PCA (Figure 4A) showed that the samples of different origins could be discriminated (with the 2D plot accounting for 90.3% of the total variance). In Figure 4B, it is possible to find boxplots for the molecular species with greater influence. Therefore, these would be the ones better best suited as candidates for origin biomarkers. When considering the 24 molecular species that contributed the most to discrimination between the three origins, we found 7 LPC, 3 PE, 3 PG, 2 PI, 2 DGDG, 1 MGDG, 1 sulfoquinovosyl diacylglycerol (SQDG), 1 MGMG, 3 DGTS, and 1 PC. These can be grouped into 8 lysolipids (7 LPC and 1 MGMG), 5 glycolipids (2 DGDG, 1 MGDG, 1 SQDG and 1 MGMG), 3 betaine lipids, and 16 phospholipids.

The differences according to growth location were also highlighted in a dendrogram with a two-dimensional hierarchical clustering (Figure 5). In this analysis, the primary split in the upper hierarchical dendrogram showed that the samples clustered independently into three groups, i.e., FR, NO, and UK. The clustering of individual phospholipid species concerning their similarity in changes of lipid expression showed that they were grouped accordingly. In the first group, several lipid species, mainly including glycolipids (SQDG) and lysoglycolipids (MGMG, DGMG, and SQMG), one PG, and two PC, which were more abundant in samples collected in France, promoted the discrimination between samples from this origin from the other ones. The second group of lipid species, including lysophospholipids (three LPE and one LPC), eight phospholipids, two SQDG, and one DGTS, which were downregulated in samples from Norway, allowed discrimination of samples from Norway and the UK.

## 4. Discussion

The impact of external conditions on the chemical composition of seaweeds has been widely addressed, and a number of growth abiotic and biotic factors have been listed as effective modulators. Therefore, different geographical locations, necessarily implying different growth conditions, may represent different environmental challenges to seaweeds which might significantly impact macroalgae composition. Sampling location has already been proven to impact macroalgae composition [39,40] in terms of fatty acids, an effect described as being as pronounced as that of seasonal changes [40].

In a first approach, we estimated the changes in elemental/biochemical composition between *S. latissima* samples from the different cultivation sites (France, Norway, and the United Kingdom), revealing some important differences. All samples displayed a remarkable inorganic content, as was expected for a brown alga [41] and was in accordance with data described in the literature for that season [11]. The elemental compositions of the samples also revealed some marked differences between geographical origins. Variability between different collection sites has been previously reported to originate differences in the presence of specific elements in *S. latissima*, namely carbon and nitrogen [42,43]. Regarding carbon, the values measured in the present work were very much in line with those found in the available literature [11,42,43], as were those for hydrogen [43]. The nitrogen/protein levels were consistent with what has been reported in literature [10,11,42]. Nitrogen/protein levels have been shown to be variable according to the collection site [42,43]. Regarding sulfur content, it was found to be remarkably higher in Norway, and, in this case, it was also somewhat higher than that previously reported in the literature for *S. latissima* samples from similar origins (wild samples from Norway and Scotland) [43,44]. 

Brown seaweeds such as *S. latissima* typically display a high content of carbohydrates, with alginate representing a major part of this fraction [11]. The values calculated fell within those expected considering earlier reports [42], while they seemed underestimated according to others [10,11]; we acknowledge that this was, in fact, an estimation. Total carbohydrate content was found to be considerably lower in Norwegian than in UK biomass (*q* = 0.001). Carbohydrate content in *S. latissima* has been reported to be modulated by growth site-specific conditions, namely salinity and nitrogen availability [42]. Moreover, in the Norwegian biomass, the increase in ash content was negatively correlated to carbohydrate levels, an effect already described in previous studies [22,43]. Lipids represent important cell components, and besides structural functions they are vital in energy and carbon storage processes, and therefore determinant for cell growth and reproduction processes. Macroalgae normally have modest lipid contents (from 0.12% to 6.73% DW), and *S. latissima* is no exception [7,10,13]. Despite this moderate lipid content, we found *S. latissima* to contain lipid species with intrinsic prospective value, while also presenting appealing nutritional features. Norwegian samples contained the highest lipid content (*q* = 0.001 vs. FR), and some abiotic factors may have been in play. A study of *S. latissima* in Denmark showed that total lipid content increased during the winter months [12]. If that is somehow indicative of an effect of temperature on lipid yield, one could assume that presumable lower water temperatures in Norway may have contributed to higher lipid content. Increasing light intensity has been shown to have a positive effect on the accumulation of lipids [45] and therefore, longer days in Norway at the time of collection (near the summer solstice) may have contributed to a higher lipid content.

The PCA score plot of fatty acid content allowed the overall clustering trends of the groups to be observed (Figure 2A) and illustrated the differences between geographical locations. If we consider that *S. latissima* cultures in Norway will tend to be subjected to lower sea temperatures given their proximity to the Artic, the fact that our results showed that biomass from that origin displayed lower content of saturated fatty acids and a higher DBI makes sense (Table 1). In fact, in macroalgae, the fatty acid composition has been reported to respond to temperature, namely by increasing the content in unsaturated fatty acids with decreasing temperature conditions [46,47,48]. In fact, a recent study pointed to temperature as the only studied physiochemical variable exerting significant influence on seaweed chemical composition of two kelp species (*S. latissima* and *Laminaria digitata*) [22]. *Saccharina latissima* has been reported to present a higher total fatty acid content during the cold winter months while exhibiting an expected tendency towards unsaturation decrease in the warmer months [14]. Interestingly, and somewhat contradictorily, another study reported January to be a time of especially lowered PUFA content, particularly of *n*-3 (EPA and DHA) and *n*-6 (linoleic acid) fatty acids [12]. Moreover, other studies [46,47,48,49,50] have reported the influence of abiotic factors, such as some that geographical circumstances may imply, on the fatty acid composition of seaweeds. Light intensity has been shown to effectively impact the fatty acid composition of seaweeds. However, the effects of light incidence in fatty acid content seem somewhat contradictory, or at least species-specific. In some cases, PUFAs have been shown to increase with photon flux density, while in others important PUFAs have been shown to be decreased [48,49,50], and in other cases fatty acid composition was not affected at all [51,52].

Lipidomic analysis by HILIC-LC-MS of total lipid extracts allowed the identification of detailed algal polar lipid profiles, highlighting geographically promoted variations. PCA analysis of the sum of the normalized peaks by polar lipid class showed a clear group separation in a two-dimensional score plot. In this approach, MGDG, PG, LPC, and DGTS totals were the most discriminating between classes. Norway and the UK displayed generally very approximate profiles per class, except for LPE and DGDG, while French biomass seemed to differ greatly from the other two sample origins in terms of lipid class profile. The amount of lysolipids were consistently higher in France compared to the other sample origins. Lysophospholipids have been traditionally considered common membrane components, but may also play important roles as second-messenger molecules regulating intracellular signaling pathways [53]. Lysophospholipids (namely LPC, LPE) are usually increased in wounded plants [54]. Temperature acclimation may also lead to increased lysophospholipid levels in plants [55]. Since northwestern France is known for having some of the highest tidal ranges, which may be reflected in increased temperature amplitude for seaweeds, this could also be one of the reasons for the increase in the polar lipid lyso forms in France samples.

Influence of light conditions may be in play when interpreting origin-promoted differences regarding the lipid classes present. A study in *Tichocarpus crinitus* (Rhodophyta) reported that low light conditions promoted an increase in SQDG, PG, and PC [45]. In *Ulva fenestrata* (Chlorophyta), low light intensity promoted increases in SQDG, PG, and MGDG [56]. These classes were all more represented in the Norwegian samples (Figure 3B); thus, low light incidence during the winter months may be reflected in polar lipid composition.

## 5. Conclusions

The results of the present study demonstrated that *S. latissima* presents a high chemical variability related to its geographical origin, revealing that the chemical phenotype is a plastic variable which can be used to discriminate between growth sites, laying the foundations for algal traceability.

Since abiotic conditions are bound to change geographically, local assessment of nutritional value and of the content in compounds of interest could be a form of valorization of regional populations and an opportunity to establish more sustainable and efficient production sites, while contributing to the marketing of more stable, reliable, and high-quality products. Ultimately, this type of study unveiling how local environmental conditions impact important characteristics of algae lipid composition may be important when selecting algae or algal strains for cultivation in a given location, especially when a set of particular components is valued. 

A regular monitoring of the chemical composition of algal biomass may be a way to gain some important insight regarding the influence of climate change on local seaweed populations, since some impacting factors are expected to change with the expected ocean acidification, temperature rise, and increase in UV radiation incidence. This may be especially relevant since it is reasonable to assume that environmentally driven changes in macroalgae content will reverberate through organisms of higher trophic levels. Moreover, the systematic screening of algae composition may be a useful resource to further predict and prevent the documented declines in populations of *S. latissima* as reported along some habitats.

## Figures and Tables

**Figure 1 biomolecules-10-00107-f001:**
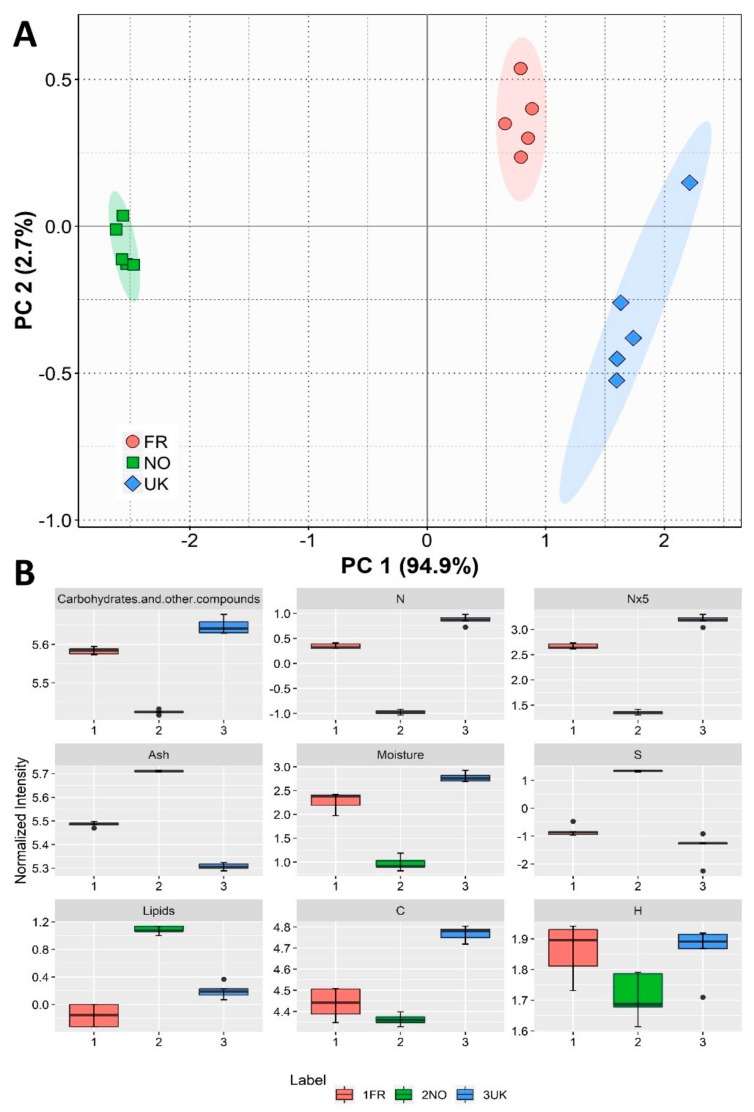
Sample origin discrimination based on biochemical composition. (**A**) Principal component analysis (PCA) score plot of elemental and biochemical traits of samples of *Saccharina latissima* from three different origins (FR—France, NO—Norway, UK—United Kingdom). (**B**) Box plots of each compositional trait used to calculate PCA, ranging from major to lesser contributors, for Component 1 of the PCA (left to right, top to bottom; C: carbon; H: hydrogen; N: nitrogen; N × 5: estimated protein content; S: sulfur).

**Figure 2 biomolecules-10-00107-f002:**
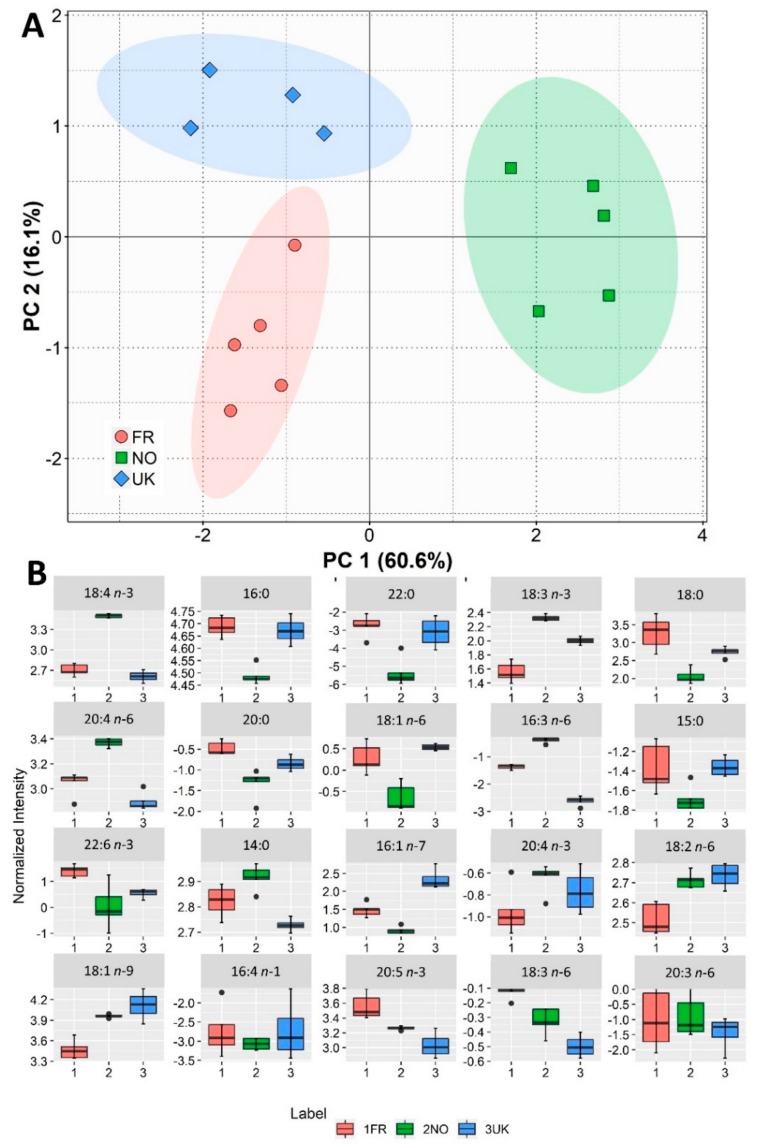
Sample origin discrimination with regard to fatty acid content. (**A**) Principal component analysis (PCA) score plot of the fatty acid profiles (in terms of relative abundance, %) for samples of *Saccharina latissima* from three different origins (FR—France, NO—Norway, UK—United Kingdom; one of the UK replicates was considered an outlier and was excluded from the analysis). (**B**) Box plots of the fatty acids used to calculate PCA ranging from major to lesser contributors for Principal Component 1 of the PCA (left to right, top to bottom).

**Figure 3 biomolecules-10-00107-f003:**
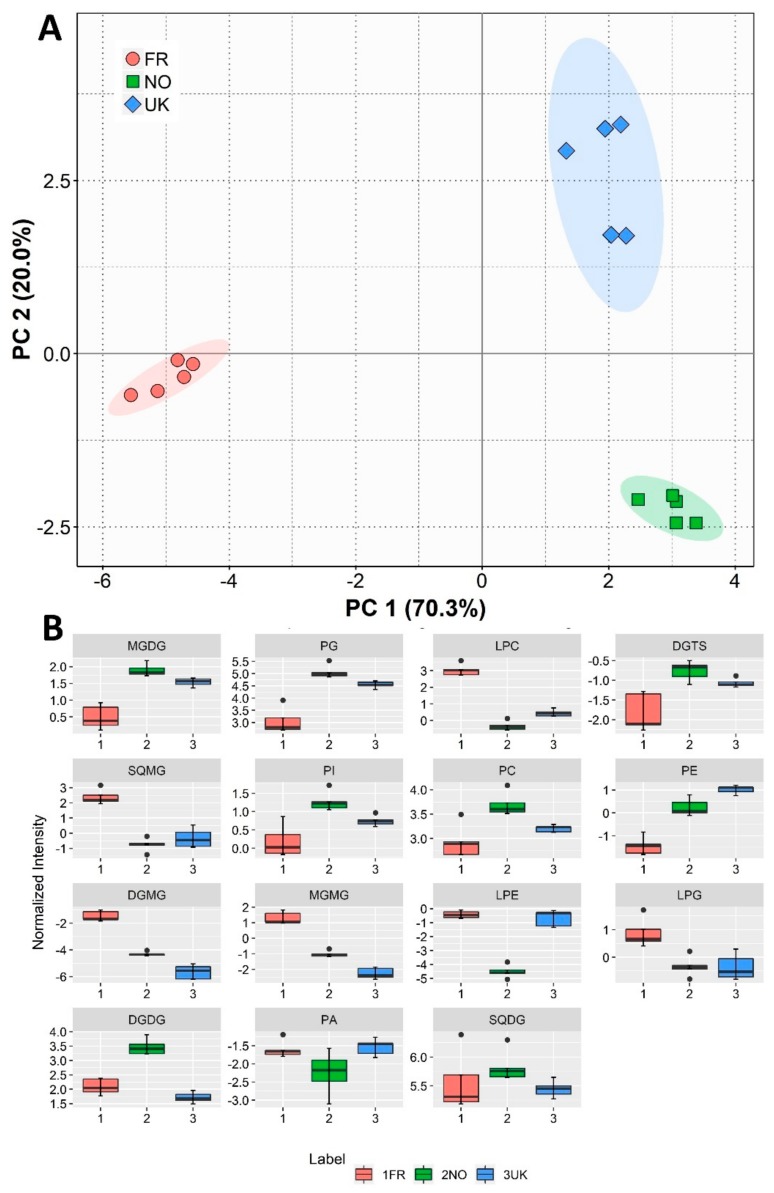
Sample origin discrimination with regard to polar lipid class content. (**A**) Principal component analysis (PCA) score plot of lipid class totals (sum of normalized peaks for each lipid class) of *Saccharina latissimi* samples from different origins (FR—France, NO—Norway, UK—United Kingdom). (**B**) Box plots for the lipid classes present in *S. latissimi* samples from different origins, ranging from major to lesser contributors for Principal Component 1 of the PCA.

**Figure 4 biomolecules-10-00107-f004:**
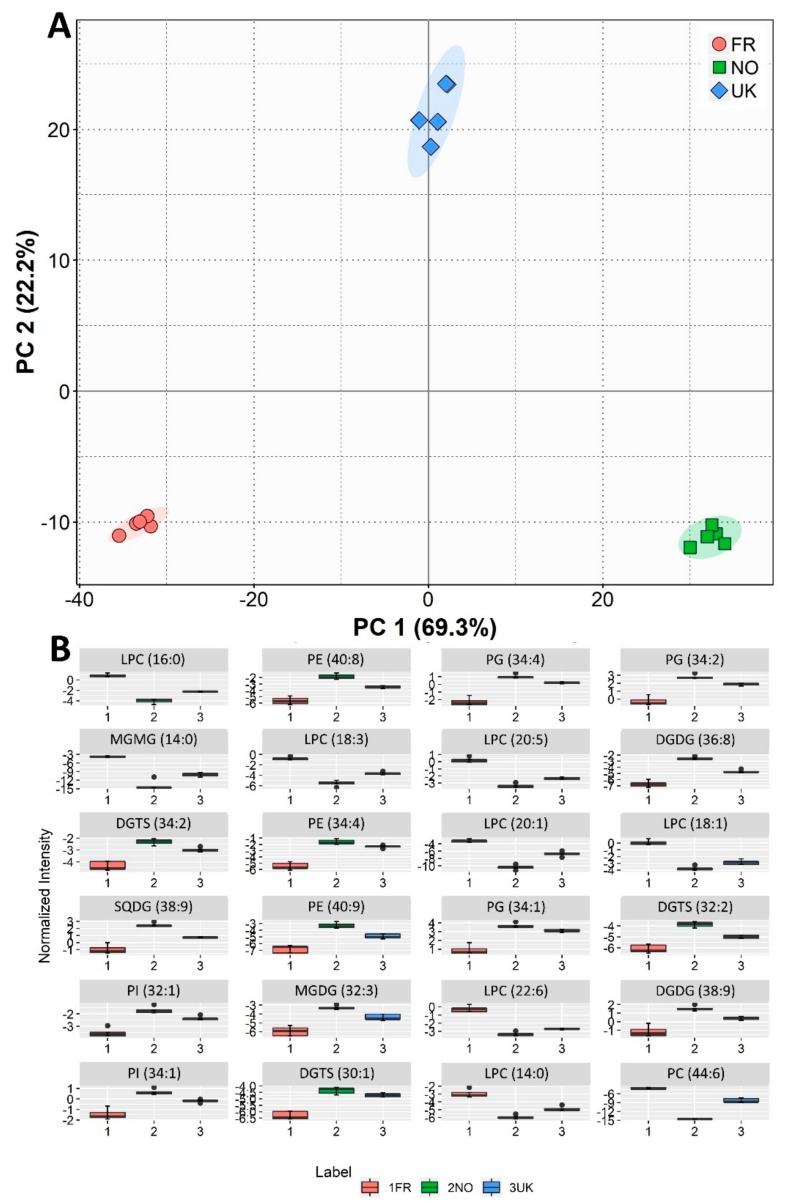
Sample origin discrimination with regard to polar lipid molecular content. (**A**) Principal component analysis (PCA) score plot of the lipid profiles in terms of the molecular polar lipid species present in samples of *Saccharina latissima* from different origins (FR—France, NO—Norway, UK—United Kingdom). (**B**) Box plots of the top 24 contributors to the first dimension of the PCA plot. Labels of the species are according to the notation AAAA (xx:i) (AAAA = lipid class; xx = total of carbon atoms in fatty acid; i = number of unsaturations).

**Figure 5 biomolecules-10-00107-f005:**
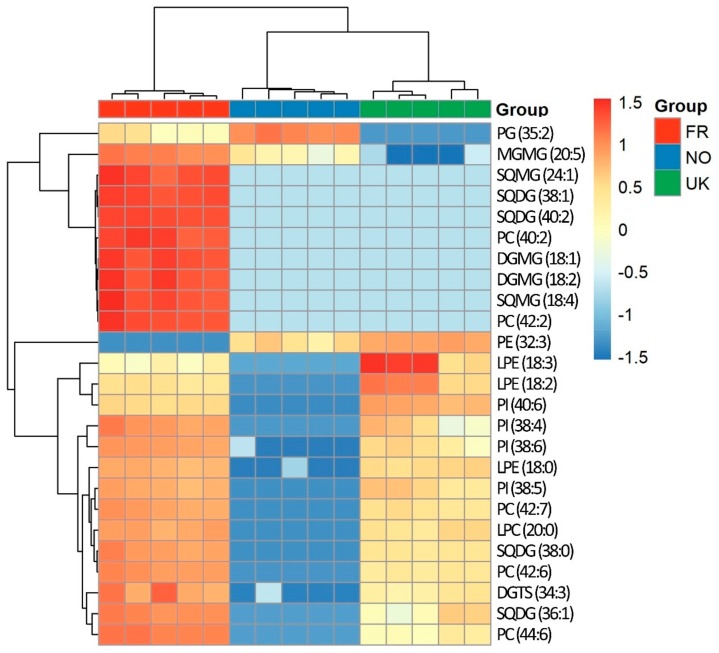
Two-dimensional hierarchical clustering heat map with the 25 most significant polar lipid discriminant species after Kruskal–Wallis statistical analysis. Levels of relative abundance are shown on the color scale, with numbers indicating the fold difference from the mean. Abbreviations: FR—France, NO—Norway, UK—United Kingdom. Labels of the species are according to the notation AAAA (xx:i) (AAAA = lipid class; xx = total of carbon atoms in fatty acid; i = number of unsaturations).

**Table 1 biomolecules-10-00107-t001:** *Saccharina latissima* fatty acid profile (%) of samples from different origins (FR—France; NO—Norway; UK—United Kingdom), derived nutritionally and functionally-relevant parameters from fatty acid profiles.

	Mean ± SEM			*q*		
Fatty Acid	FR	NO	UK	FR vs. NO	FR vs. UK	NO vs. UK
**12:0**	0.07 ± 0.05	0.00 ± 0.00	0.00 ± 0.00			
**14:0**	7.08 ± 0.13	7.55 ± 0.11	6.75 ± 0.14			0.009
**15:0**	0.39 ± 0.03	0.30 ± 0.01	0.37 ± 0.02			
**16:0**	25.79 ± 0.32	22.46 ± 0.26	24.99 ± 0.65	0.014		0.036
**16:1 *n*-7**	2.82 ± 0.17	1.90 ± 0.06	4.52 ± 0.76			0.007
**16:3 *n*-6**	0.39 ± 0.01	0.76 ± 0.02	0.16 ± 0.01			0.001
**16:4 *n*-1**	0.16 ± 0.04	0.12 ± 0.01	0.12 ± 0.01			
**18:0**	10.07 ± 1.37	4.19 ± 0.28	6.11 ± 0.66	0.007		
**18:1 *n*-9**	11.10 ± 0.49	15.59 ± 0.12	17.63 ± 1.03		0.004	
**18:1 *n*-6**	1.24 ± 0.14	0.65 ± 0.07	1.45 ± 0.04	0.029		0.013
**18:2 *n*-6**	5.73 ± 0.14	6.56 ± 0.08	6.57 ± 0.16	0.013	0.013	
**18:3 *n*-6**	0.93 ± 0.00	0.80 ± 0.02	0.72 ± 0.02		0.004	
**18:3 *n*-3**	2.95 ± 0.13	5.00 ± 0.07	4.18 ± 0.19	0.002		
**18:4 *n*-3**	6.52 ± 0.17	11.32 ± 0.09	6.11 ± 0.18	0.043		0.007
**20:0**	0.72 ± 0.04	0.41 ± 0.04	0.54 ± 0.03	0.002		
**20:3 *n*-6**	0.58 ± 0.16	0.58 ± 0.13	0.45 ± 0.08			
**20:4 *n*-6**	8.51 ± 0.05	10.34 ± 0.11	7.89 ± 0.47			0.006
**20:4 *n*-3**	0.52 ± 0.04	0.64 ± 0.03	0.60 ± 0.04			
**20:5 *n*-3**	11.83 ± 0.64	9.60 ± 0.07	8.42 ± 0.44		0.003	
**22:0**	0.16 ± 0.03	0.02 ± 0.00	0.11 ± 0.04	0.019		
**22:6 *n*-3**	2.68 ± 0.19	1.19 ± 0.33	1.64 ± 0.18	0.018		
**Indexes**						
**SFA**	44.28 ± 1.71	34.95 ± 0.58	38.88 ± 1.10	0.004		
**MUFA**	15.16 ± 0.67	18.14 ± 0.13	23.48 ± 1.01		0.001	
**PUFA**	40.56 ± 1.37	46.91 ± 0.52	37.64 ± 1.81			0.007
**PUFA *n*-3**	24.50 ± 1.13	27.75 ± 0.39	21.56 ± 1.29			0.007
**PUFA *n*-6**	17.13 ± 0.40	19.69 ± 0.22	17.25 ± 0.51	0.016		0.016
***n*-6/*n*-3**	0.70 ± 0.02	0.71 ± 0.01	0.81 ± 0.03			
**Sat/Unsat**	0.80 ± 0.05	0.54 ± 0.01	0.64 ± 0.03	0.004		
**ACL**	17.67 ± 0.03	17.66 ± 0.02	17.55 ± 0.04			
**DBI**	178.3 ± 6.33	197.5 ± 2.43	167.0 ± 6.91			0.018
**PI**	170.1 ± 6.78	178.1 ± 2.82	143.6 ± 8.20			0.033

Values presented are means ± standard error of mean for five replicates. Statistical differences are presented as others vs. France samples (noted as * symbols), and UK vs. Norway (noted as # symbols). Only differences with *q* value < 0.05 were considered statistically significant. ACL: average chain length; DBI: double bond index; PI: peroxidizability index.

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
