# Peer review of "The Unique Lipidomic Signatures of Saccharina latissima Can Be Used to Pinpoint Their Geographic Origin"

_biomolecules, 2020, doi:10.3390/biom10010107_

Round 1

Reviewer 1 Report

A paper entitled "Saccharina latissima from different culture locations display unique lipidomic signatures that can be used to trace their geographic origin" submitted to Biomolecules for reviewing and publication. The paper is well written. Novelty in the chemistry and biology are interesting to attract reader's interesting. I recommend that this submission is acceptable for publication with its present for after minor revision.

Please reduce the sections "introduction" and "Materials and Methods". In general, the authors can cite the reference to replace the unnecessary details.

Author Response

"Please reduce the sections "introduction" and "Materials and Methods". In general, the authors can cite the reference to replace the unnecessary details."

We reduced the "Introduction" section by eliminating some sentences, namely lines 63, 70 and 80 of the new submitted manuscript version with track changes.

The "Materials and Methods" were significantly reduced and reshaped, namely the "Lipid extraction", the "Fatty acid analysis using gas chromatography – mass spectrometry (GC-MS)", the "Hydrophilic interaction liquid chromatography–mass spectrometry (HILIC-LC-MS)" and the "Statistical analysis" sections, essentially by referring to previous references from works in our lab.

Reviewer 2 Report

In the present manuscript, the authors show the lipidomic difference between an algae collected in three different geographical regions. I just find two minor considerations.

Authors need to mention if the sampled of the algae, were done using the complete algae or just a part of it. 

Line 146. The column was 30 m length?

Author Response

"Authors need to mention if the sampled of the algae, were done using the complete algae or just a part of it"

We clarified this issue by including the sentence "Whole seaweed portions from each location were freeze-dried, grinded with a mortar and pestle and kept at -80 ºC before analysis. For each localization, five different random portions of the biomass were used." at the end of the "Sampling" section of the "Materials and Methods".

"Line 146. The column was 30 m length?"

The lenghth is actually 30 m, but we decided to refer to a citation of a previous work anyways.